# FluidRegNet: Longitudinal registration of retinal OCT images with new pathological fluids

**Julia Andresen**[1]                   J.andresen@uni-luebeck.de
**Jan Ehrhardt**[1,2]                   Jan.Ehrhardt@dfki.de
**Claus von der Burchard**[3]           Claus.vonderBurchard@uksh.de
**Ayse Tatli**[3]                     Ayse.Tatli@uksh.de
**Johann Roider**[3]                  Johann.Roider@uksh.de
**Heinz Handels**[1,2]                 Heinz.Handels@dfki.de
**Hristina Uzunova**[2]                Hristina.Uzunova@dfki.de

[1] *University of Lübeck, Institute of Medical Informatics, Lübeck*
[2] *German Research Center for Artificial Intelligence (DFKI), Lübeck*
[3] *Christian-Albrecht University of Kiel, Department of Ophthalmology, Kiel*

**Editors:** Accepted for publication at MIDL 2024

## Abstract

Eye diseases such as the chronic central serous chorioretinopathy are characterized by fluid deposits that alter the retina and impair vision. These fluids occur at irregular intervals and may dissolve spontaneously or thanks to treatment. Accurately capturing this behavior within an image registration framework is challenging due to the resulting prominent tissue deformations and missing image correspondences between visits. This paper presents FluidRegNet, a convolutional neural network for the registration of successive optical coherence tomography images of the retina. The correspondence between time points is established by predicting the position of the origin of the fluids by creating a fluid seed in the form of sparse intensity offsets in the moving image and registering the fluid seed to the affected area in the follow-up image. We show that this leads to deformation fields that more accurately reflect the actual dynamics of retinal fluid growth compared to other image registration methods. In addition, the network outputs are used for unsupervised fluid segmentation.

**Keywords:** Image registration, unsupervised deep learning, optical coherence tomography, retinal fluids, image segmentation.

## 1. Introduction

The eye disease central serous chorioretinopathy (CSCR) is characterized by the formation of fluid deposits below and sometimes within the retina, which severely impair the patient's vision. Although these fluids are clearly visible on optical coherence tomography (OCT) images, the underlying pathomechanisms are still poorly understood (Pfau et al., 2021). However, it is known that the fluid distribution can change considerably within short time, which means that OCT images from successive examinations can have a very different shape and appearance. Nevertheless, the correct quantification of fluid development is a crucial step in understanding disease development and progression. A longitudinal image registration framework establishing spatial correspondence between images from different

time points is, therefore, a key for the computer-aided assessment of retinal OCT time-series, and can be used for e.g. volume change tracking, progress control or automatic segmentation by only requiring the ground truth annotation of the first time point.

Yet, the nature of fluids in the retina can be extremely varying, showing different biophysical properties. For example, intraretinal fluids (IRF) cause local displacements of the normal retinal tissue, while subretinal fluids (SRF) lift the entire overlying retina. Additionally, different treatments are able to reduce or even completely dissolve the fluid deposits, that, however, can repeatedly emerge in the further course of image acquisition. This dynamic of emerging and dissolving fluids does not allow for conventional longitudinal registration approaches, due to the missing correspondences between time points, requiring for a specialized approach that can capture the dynamics of retinal fluid growth in CSCR.

The longitudinal registration of OCT images has been studied in the literature primarily for non-pathological images, e.g. (Niemeijer et al., 2009; Lang et al., 2016; Lee et al., 2017; Gong et al., 2019). For images with pathologies, existing image registration methods (Wei et al., 2017; Pan et al., 2019, 2020; Andresen et al., 2022a) can successfully be used to model size changes of existing fluids, yet, they usually fail to accurately capture the nature of the displacements caused by newly emerging fluid deposits. Often a smearing of the normal retinal tissue over the fluid area is observed (Andresen et al., 2022a). Furthermore, most methods require given segmentations of retinal layers (Wei et al., 2017; Pan et al., 2019, 2020), which need to be acquired through laborious and time-consuming manual work.

A common approach to tackle the problem of missing correspondences is to model the causing structures (here: new lesions) directly and integrate them into the registration process as prior knowledge, as in (Shin et al., 2018; Uzunova et al., 2019; Andresen et al., 2022b). However, such modeling approaches often do not account for the healthy tissue displacement caused by the pathology, which, in the case of retinal OCTs, might be severe. An attempt to implicitly model these fluid-induced deformations is given by (Uzunova et al., 2022), however, this method does not explicitly model the pathological displacement, and, furthermore, requires ground truth fluid segmentations.

Another concept to model emerging pathological structures during image registration is to account for their differing appearance by using metamorphosis models. They can jointly estimate appearance and shape differences between images and have been shown to be suitable for the registration of pathological images. To the best of our knowledge, these methods either consider appearance differences for the entire imaged tissue (Meng et al., 2022) or require exact ground truth pathology segmentations to model only the appearance differences in the pathology regions (Niethammer et al., 2011; Maillard et al., 2022; François et al., 2022; Joshi and Hong, 2023; Wang et al., 2023).

Inspired by metamorphosis methods, in this work, we present a deep learning-based approach for the longitudinal registration of OCT images that is capable of capturing the dynamics of retinal fluid growth in a fully unsupervised manner. The proposed fluid registration network, FluidRegNet, is not only able to model existing growing and shrinking lesions by predicting spatial deformations, but also newly emerging fluids can be represented using a novel sparse appearance seed approach. This leads to more realistic displacements of the healthy and pathological tissue compared to conventional registration approaches. Furthermore, the proposed method does not require any ground truth pathology labels for training and can also be used for an unsupervised segmentation of retinal fluids.

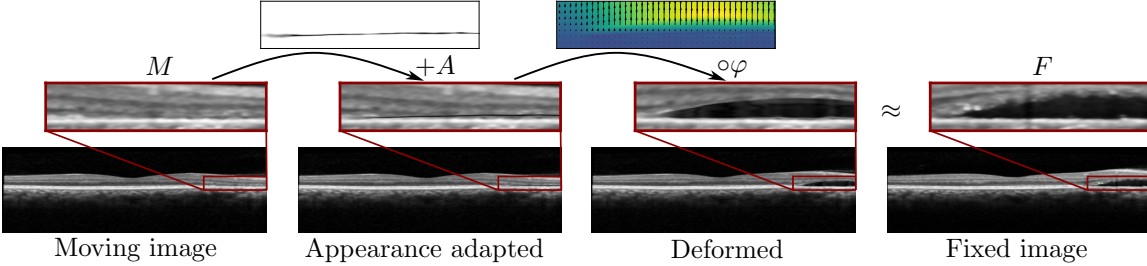

$M$  $+A$  $\circ\varphi$  $F$

$\approx$

Moving image  Appearance adapted  Deformed  Fixed image

Figure 1: An overview of the proposed sparse appearance seed registration framework. A sparse fluid seed $A$ is inserted into the moving image $M$ before applying the deformation field $\varphi$, so that the deformed image corresponds to the reference image in appearance and morphology.

## 2. Methods

The goal of our registration method is to geometrically align OCT images from different time points featuring realistic, fluid-induced deformations. While growing and shrinking lesions can be captured by the resulting deformation field, newly emerging or dissolving lesions cannot be solely modeled by spatial deformations, as there are no corresponding lesion pixels in the reference image. In this work, we propose to explicitly and sophisticatedly tackle the clinically relevant problem of newly emerged pathological fluids as a major clinical biomarker. We present FluidRegNet, a registration framework that addresses the issues arising from the frequently changing clinical manifestations of fluids in CSCR. The intuition behind the presented approach is to mimic the onset of fluid formation in the moving image by inserting so-called lesion appearance seeds: minimal regions of changed intensities which can be then stretched to the fluid region by a suitable deformation field (Fig. 1). Since we assume that the deformations inside and outside fluid deposits are different, a masked regularizer loss is used, that also implicitly tackles the problem of dissolving fluids.

### 2.1. Sparse appearance seed registration framework

Inspired by metamorphic approaches, our method assumes that a follow-up image $F$ can be reconstructed from a baseline moving image $M$ by applying an appearance offset map $A$ and a deformation field $\varphi$, such that $F \approx (M + A) \circ \varphi$. Since our scenario considers follow-up images of the same patient, the appearance offset would typically be small, however, here we model $A$ such that it represents the seed of an emerging retinal lesion, thus, $A$ needs to be sparse. These assumptions yield the following objective:

$$\mathcal{L} = \mathcal{L}_{\text{Dist}}\left(F, (M + A) \circ \varphi\right) + \alpha\mathcal{L}_{\text{Reg}}(\varphi) + \beta\mathcal{L}_{\text{Sparse}}(A), \tag{1}$$

where $\mathcal{L}_{\text{Dist}}$ is the normalized cross correlation image distance loss that has successfully been applied for OCT image registration before (Pan and Chen, 2023), $\mathcal{L}_{\text{Reg}}$ serves to regularize the deformation field and $\mathcal{L}_{\text{Sparse}}$ enforces sparsity of the appearance offsets.

The regularization of the deformation field is crucial in the presented method, thus, the following multi-component regularization loss is used:

$$\mathcal{L}_{\text{Reg}}\left(\varphi\right) = \mathcal{L}_{\text{Dice}}\left(S_F^{\text{retina}}, S_M^{\text{retina}} \circ \varphi\right) + \gamma\mathcal{L}_{\text{Jac}}\left(\varphi\right) + \delta\mathcal{L}_{\text{Diff}}\left(\varphi\right). \tag{2}$$

Here, $\mathcal{L}_{\text{Dice}}$ is the Dice loss between the retina segmentation of the follow-up image $\text{S}_{\text{F}}^{\text{retina}}$ and the warped retina segmentation of the moving image $\text{S}_{\text{M}}^{\text{retina}}$, giving weak guidance to the registration task and ensuring good overlap of the entire imaged tissue. Since the fluid volume can change very strongly and to different degrees for each image pair, we use a masked diffusion loss $\mathcal{L}_{\text{Diff}}$ that ensures smooth displacement fields for the normal retinal tissue but allows arbitrary deformations in fluid regions. For this purpose, rough lesion segmentations are generated by binary thresholding of moving and fixed images. The resulting masks are dilated and combined to form a pseudo-lesion mask $\Lambda$. The diffusion loss is then given by $\mathcal{L}_{\text{Diff}}(\varphi) = \frac{1}{|\Omega \backslash \Lambda|} \sum_{\boldsymbol{x} \in \Omega \backslash \Lambda} \sum_{i=1}^{2} \|\nabla \varphi_i(\boldsymbol{x})\|_2^2$, where $\Omega$ is the set of all image pixels. Still, foldings are avoided both inside and outside fluids by penalizing negative values of the Jacobian determinant of the deformation field $J_\varphi$ using $\mathcal{L}_{\text{Jac}} = \max(0, -|J_\varphi|)^2$.

The final part of Eq. 1 is the appearance sparsity loss:

$$\mathcal{L}_{\text{Sparse}}(A) = \sum_{\boldsymbol{x} \in \Omega} |A(\boldsymbol{x})| + \eta \max(0, A(\boldsymbol{x}))^2 \tag{3}$$

that favors small and, due to the second term, negative values of $A$, which effectively leads to sparse appearance maps reflecting the low intensities of fluids in OCT images.

## 2.2. Architecture and implementation details

The architecture of FluidRegNet builds on U-Net with two separate output heads for deformation field and appearance offset map. On each level, two convolutional layers with kernel size 3 followed by batch normalization and leaky ReLU activation are used. Two inputs are given to the network: a two-channel image consisting of the moving and fixed images; and a one channel difference image $M - F$. The inputs are first processed by two separate input blocks and then combined by concatenation. For exact architecture, see appendix (Fig. 4).

As observed in previous work (Andresen et al., 2022a), training from scratch with a multicomponent loss function, such as Eq. (1), can degrade the network's performance due to the complicated entanglement of shape and appearance. We, therefore, follow a three-step training scheme. First, the network is pre-trained for 200 epochs only considering the deformation output of the CNN. Second, a warm-up step is used throughout the next 50 epochs, in which only the layers generating the sparse appearance map are trained while the remaining layer weights are frozen. Finally, the entire network is fine-tuned for 450 more epochs. Network training is performed with a batch size of 10 on a single NVIDIA Tesla V100 GPU with 32 GBs of RAM using Adam optimization, an initial learning rate of $1\text{e}^{-4}$ and a learning rate decay of 0.8 after every tenth of the epochs. The weight parameters $\alpha$, $\beta$, $\gamma$, $\delta$ and $\eta$ are set to 1, $3\text{e}^{-4}$, 1000, 1 and 210, accordingly.

## 3. Experiments and Results

### 3.1. Data

We develop our model on 369 OCT images from 61 eyes of 33 CSCR patients acquired longitudinally with a Spectralis OCT device. Manual annotations of the retina, IRF, SRF and pigment epithelial detachment (PED) have been generated by medical experts for 19 patients. This results in 163 manually annotated image volumes, 105 of which contain fluids.

For the remaining patients, only retina segmentations are given. The follow-up period ranges from two months to eleven years, with two to 17 images per eye (see appendix, Table 3 for details). All images are sized $496 \times 512 \times 25$ voxels with a field of view of $2 \times 6 \times 6$ mm$^3$. The pre-processing of the images features flattening at the Bruch's membrane, denoising using a guided filter with radius 1 (He and Sun, 2015) and normalization to the intensity range $[0, 1]$. All experiments are carried out on 2D B-scans with five-fold cross-validation using 80% of patients for training and 20% for testing. Results are averaged over all test images.

## 3.2. Registration accuracy

To evaluate registration accuracy, each image is aligned to its subsequent follow-up and results are compared against image registration benchmarks ANTs SyN (Avants et al., 2008) and VoxelMorph (Balakrishnan et al., 2019). For fair comparison, FluidRegNet is also trained as a "classical" registration framework, using the diffusion regularizer on the entire image domain and omitting the appearance offset branch. VoxelMorph and the classical FluidRegNet are trained for 700 epochs and with the same loss function. ANTs SyN is performed with cross correlation image similarity metric, four resolution levels, no initial affine transformation and the default settings for all other parameters.

Results are reported in Table 1 for the manually annotated images, showing absolute symmetric surface distance (ASSD) and 95% Hausdorff distance (HD) of the inner limiting membrane (ILM). Deformation field regularity is assessed by the average number of pixels with $|J_\varphi| < 0$. Additionally, we show that our region-based regularizer in combination with sparse appearance offsets leads to more realistic deformations by evaluating the volume change $|1 - |J_\varphi||$ separately in normal tissue and fluid regions. It is expected that the volume changes in the fluid regions will be larger than in the retinal tissue because the tissue is displaced by the fluid changes. Results show that FluidRegNet outperforms VoxelMorph in terms of registration accuracy. While the regularity of the deformation fields of VoxelMorph is best, it impairs its ability to capture large deformations (Fig. 2). Compared to SyN, FluidRegNet produces more regular deformation fields, while SyN aligns the ILM more accurately. The deformations resulting from SyN however are unrealistic for image registration pairs with newly forming fluids (Fig. 2). In contrast, FluidRegNet provides plausible deformations that are mainly large in the area of the pathologies, while the healthy tissue is displaced accordingly. Despite the different behavior inside and outside fluids, the resulting deformation fields contain very few inversions, assessed by an average of 54.17 negative Jacobian determinant pixels, corresponding to only 0.02% of all pixels.

## 3.3. Unsupervised segmentation of new fluids

In this experiment, two assumptions are made to achieve an unsupervised segmentation of newly emerged retinal fluids: New fluids correspond to image regions with, firstly, a strong increase in volume and, secondly, large appearance offsets. Thus, the set of segmented new lesion pixels is given by $\mathcal{S} = \{\boldsymbol{x} \in \Omega | \varphi(A(\boldsymbol{x})) < \tau_A \vee |J_\varphi(\boldsymbol{x})| > \tau_\varphi\}$. The thresholds $\tau_A$ and $\tau_\varphi$ are found using grid search and set to -0.017 and 4.7 respectively in order to achieve the highest segmentation accuracy while providing good detection results. For comparison, two unsupervised anomaly segmentation approaches are applied: Natural synthetic anomalies (NSA, (Schlüter et al., 2022)), which uses Poisson image inpainting to generate synthetic

Table 1: Registration results for FluidRegNet (FRN) compared to VoxelMorph (VXM, (Balakrishnan et al., 2019)) and symmetric image normalization (SyN, (Avants et al., 2008)). Used metrics: absolute symmetric surface distance (ASSD) and 95% Hausdorff distance (HD) to the inner limiting membrane (ILM) in micrometers; and various evaluations of the displacement field Jacobian $J_\varphi$.

| Method | ASSD ILM $\downarrow$ | HD ILM $\downarrow$ | $|J_\varphi| \leq 0 \downarrow$ | $|1-|J_\varphi||_{\text{healthy}} \downarrow$ | $|1-|J_\varphi||_{\text{fluid}} \uparrow$ |
|---|---|---|---|---|---|
| Before | $10.44 \pm 22.64$ | $36.88 \pm 55.10$ | - | - | - |
| VXM | $9.88 \pm 19.21$ | $27.84 \pm 50.52$ | $\mathbf{29.30} \pm 70.06$ | $\mathbf{0.13} \pm 0.05$ | $0.28 \pm 0.13$ |
| SyN | $\mathbf{5.45} \pm 9.59$ | $\mathbf{16.12} \pm 36.31$ | $601.30 \pm 973.71$ | $0.19 \pm 0.08$ | $0.44 \pm 0.20$ |
| FRN$_{\text{classic}}$ | $8.15 \pm 15.67$ | $23.51 \pm 43.47$ | $49.24 \pm 115.12$ | $0.14 \pm 0.07$ | $0.36 \pm 0.17$ |
| FRN | $7.78 \pm 15.18$ | $22.01 \pm 40.12$ | $54.17 \pm 145.36$ | $0.14 \pm 0.06$ | $\mathbf{0.95} \pm 0.77$ |

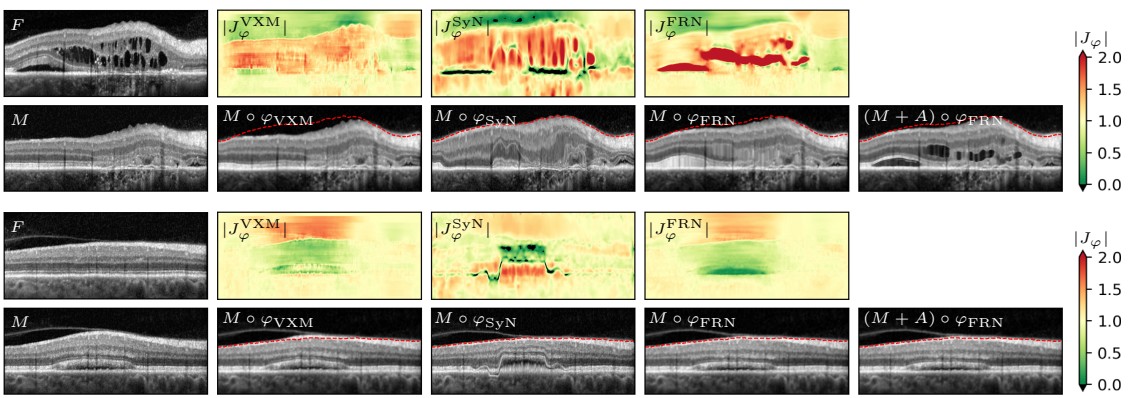

Figure 2: Registration results for VoxelMorph (VXM), SyN and FluidRegNet (FRN) for images with missing correspondences due to fluids being present in only one of the two time points. Shown are moving and fixed images, the Jacobian determinant of the deformation fields $J_\varphi$ and the deformed moving images with the ILM of the fixed image overlaid in red. Images are shown with isotropic pixel spacing.

anomalies, a core component of the Medical Out-of-Distribution Analysis Challenge[1] winner approach (Baugh et al., 2023); and f-AnoGAN (Schlegl et al., 2019), a GAN-based approach developed specifically for OCT images and trained on healthy images only (here, an internal dataset of 50 healthy subjects).

The results for unsupervised detection and segmentation of new fluids are shown in Table 2. To evaluate the segmentations, the mean Dice Score (DSC) between the predicted and the ground truth segmentations of new lesions is reported. Also, the detection rate of new lesion cases is reported, classifying each image with DSC > 0 as correctly detected. Since the generated segmentations do not distinguish between IRF, SRF and PED, we first present results for all pathologies combined in one binary segmentation. DSCs per pathology type are then reported by calculating the average DSC between the ground truth per lesion type and each predicted new lesion overlapping with the ground truth.

The results show that FluidRegNet reliably detects newly formed IRF (94.44%) and SRF (86.22%). Newly developed PED, however, is detected in only 27.59% of cases (examples in

---

1. http://medicalood.dkfz.de/web/

Table 2: New fluids segmentation results. Metrics for all fluids, IRF, SRF and PED: detection rate (sens) and Dice score averaged over all lesion detected images ($\mathrm{DSC_{det}}$).

| Method | All | | IRF | | SRF | | PED | |
|---|---|---|---|---|---|---|---|---|
| | sens | $\mathrm{DSC_{det}}$ | sens | $\mathrm{DSC_{det}}$ | sens | $\mathrm{DSC_{det}}$ | sens | $\mathrm{DSC_{det}}$ |
| NSA | **236/275** | 0.24 | **34/36** | 0.23 | **184/196** | 0.26 | **29/58** | 0.19 |
| f-AnoGAN | 181/275 | 0.27 | 29/36 | 0.22 | 140/196 | 0.29 | 23/58 | 0.23 |
| FRN | 215/275 | **0.58** | **34/36** | **0.59** | 172/196 | **0.67** | 18/58 | **0.52** |

the appendix, Fig. 5). It can be noted that new PEDs are indeed the most challenging for all methods, since they are majorly underrepresented in the dataset (only 58 B-scans overall) and generally lie in the same area as SRFs. Compared to our CNN, NSA performs a better detection for all lesion types. In terms of segmentation accuracy, however, FluidRegNet achieves significantly better results and also outperforms f-AnoGAN in all metrics.

### 3.4. Unsupervised chronological segmentation

Assuming that the patients' initial visits are segmented manually, we use FluidRegNet to chronologically segment fluids for the successive visits. Each longitudinal patient image $I_i$, with $i \in [0, t-1]$ and $t$ the last acquisition time point, is registered to its subsequent $I_{i+1}$ using the deformation field $\varphi_i$. Starting from the segmented initial examination $S_0$, the subsequent segmentations are generated for each patient as $S_{i+1} = S_i \circ \varphi$. Fluids that reappear or are newly emerging in the course of treatment are modeled as described in the previous section and added to the segmentation of the previous time point. Similarly, segmentations of dissolving pathologies are removed if $|J_{\varphi_i}| < 0.3$ for at least 90% of the lesion pixels or the maximum thickness of the remaining fluid is less than five pixels. For comparison, NSA and f-AnoGAN are applied for the segmentation of each time point individually.

Fig. 3 shows quantitative and qualitative results for the chronological fluid segmentation. For images with fluids in the manual ground truth, the DSC is calculated, whereas for images with no ground truth fluids, the average number of false positive pixels is reported. Requiring only one segmented start time point, FluidRegNet provides the best segmentation accuracy and the smallest amount of false positives compared to fully unsupervised anomaly detection methods. FluidRegNet can, thus, be used to accurately estimate fluid distribution and volume in time series data in an unsupervised manner.

### 4. Discussion

In this paper, we presented FluidRegNet, an unsupervised registration framework for longitudinal OCT images containing pathological fluids. The presented approach overcomes the problem of missing correspondences for the registration of images with newly emerging fluids, by introducing a sparse appearance seed as an attempt to model the onset of pathological structures before applying the deformation field. Combined with a sophisticated regularization scheme, FluidRegNet provides more realistic, fluid-aware deformations and is able to handle severe appearance and shape differences between images. Our experiments show, that next to an improved registration accuracy, the presented method can

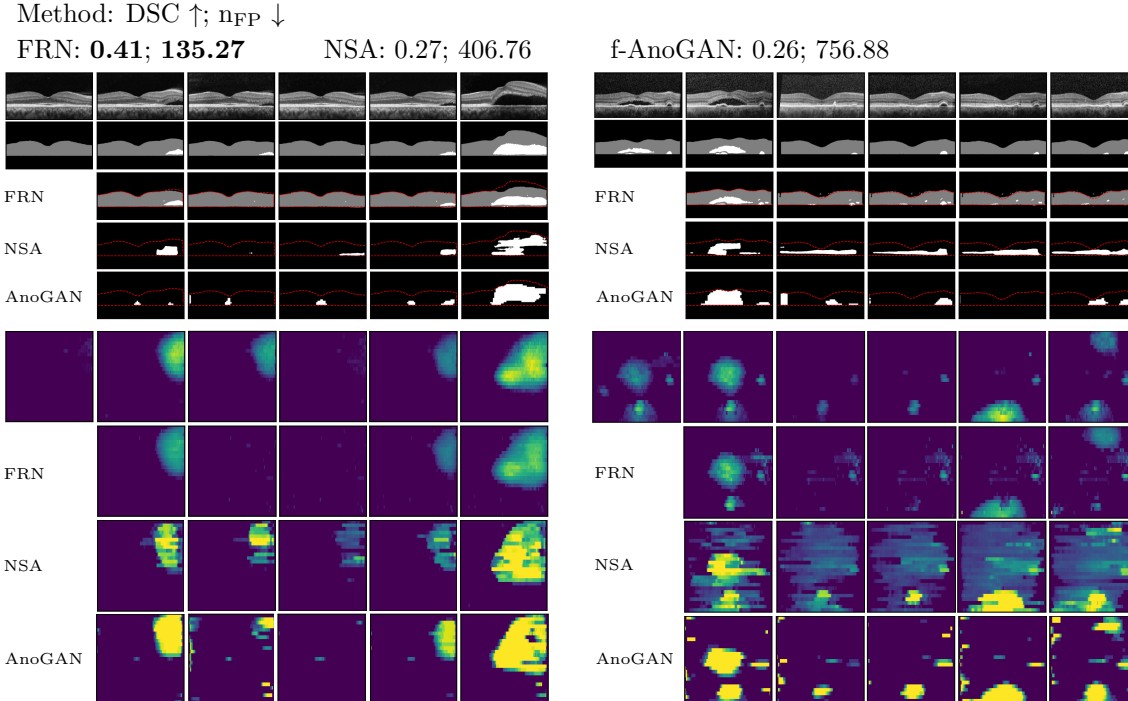

Figure 3: Chronological segmentation results for FluidRegNet (FRN), NSA and f-AnoGAN. Quantitative results are shown above (DSC and number of false positive pixels n_FP), while the qualitative results for two patients are shown below. The figure shows from top to bottom: central OCT B-scans, ground truth segmentations, predicted segmentations (per method) and the en-face projections of segmentations in the same order. Isotropic pixel spacing is used for better visualisation.

also be reliably applied for unsupervised anomaly detection, and shows improved or on-par performance to state-of-the-art methods in both tasks.

Even though a substantial improvement is achieved in this area, like most learning-based image registration methods, FluidRegNet has difficulties in capturing very large deformations, which we plan to address in future research. We also plan to explicitly address the problem of dissolving fluids and capture them in our framework in an improved manner. For clinical use, further tests need to be carried out on the performance of FluidRegNet in diseases with more complicated fluid patterns such as age-related macular degeneration. In addition, cases in which different fluid types occur in the same image regions should be investigated in more detail. The choice of multiple weighting parameters in the loss function is another drawback of our method. We suggest beginning with a classical registration optimizing hyperparameters for image distance and spatial regularization only before introducing appearance offsets.

FluidRegNet provides a valuable tool for longitudinal registration of OCT images that captures the actual dynamics of retinal fluid growth better than other methods. Its application for unsupervised segmentation of time-series images could significantly reduce the cost of their tedious manual annotation. Finally, the realistic nature of our deformation fields enables detailed visualization of regressive and progressive areas in the eye.

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

## Appendix A. Network Training Details

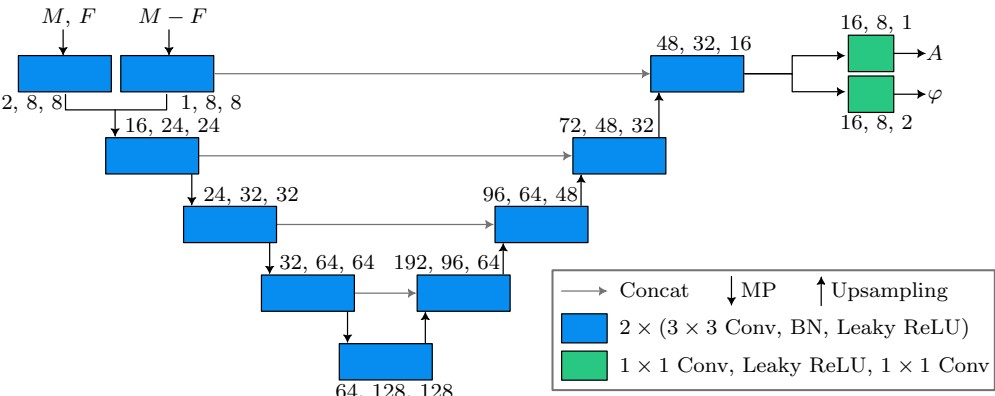

Figure 4: Network architecture of FluidRegNet. Input are corresponding B-Scans of fixed and moving images. Outputs are the appearance offset map $A$ and the deformation field $\varphi$. Maximum pooling (MP) with stride 2 is used in the encoding and bilinear upsampling in the decoding path. BN stands for batch normalization. Numbers above (below) boxes indicate the number of input features and the number of feature maps resulting from the convolutional layers.

Table 3: Description of the longitudinal OCT dataset used. For each CSCR patient, the number of (segmented) images per eye (#L, #R), the fluids observed in the respective eye (0: None, 1: SRF, 2: IRF, 3: PED) and the follow-up time (FUT) in months between first and last visit is given.

| ID | #L | Path. | #R | Path. | FUT | ID | #L | Path. | #R | Path. | FUT |
|----|------|---------|--------|---------|-----|----|--------|---------|---------|---------|-----|
| 01 | 6 (6) | 1, 2 | 7 (7) | 1, 2 | 32 | 18 | 6 (1) | 1, 2 | 6 (1) | 0 | 33 |
| 02 | 17 (0) | - | 17 (0) | - | 90 | 19 | 4 (0) | - | 4 (0) | - | 9 |
| 03 | 7 (7) | 1, 2 | 7 (7) | 0 | 97 | 20 | 7 (2) | 1 | 9 (2) | 1 | 77 |
| 04 | 3 (3) | 0 | 3 (3) | 0 | 70 | 21 | 4 (0) | - | 10 (0) | - | 41 |
| 05 | 0 (0) | - | 17 (0) | - | 134 | 22 | 8 (8) | 1, 2, 3 | 8 (8) | 1, 2, 3 | 119 |
| 06 | 2 (2) | 0 | 2 (2) | 1 | 93 | 23 | 10 (0) | - | 10 (0) | - | 117 |
| 07 | 2 (0) | - | 2 (0) | - | 115 | 24 | 3 (0) | - | 3 (0) | - | 74 |
| 08 | 0 (0) | - | 5 (5) | 0 | 74 | 25 | 6 (5) | 1, 2 | 6 (3) | 1 | 78 |
| 09 | 5 (5) | 1, 2 | 3 (3) | 1, 2, 3 | 41 | 26 | 1 (0) | - | 2 (0) | - | 12 |
| 10 | 4 (0) | - | 6 (0) | - | 28 | 27 | 7 (0) | - | 15 (0) | - | 130 |
| 11 | 8 (8) | 0 | 8 (8) | 1 | 28 | 28 | 2 (0) | - | 2 (0) | - | 2 |
| 12 | 0 (0) | - | 2 (0) | - | 3 | 29 | 3 (0) | - | 3 (0) | - | 8 |
| 13 | 6 (1) | 1 | 6 (0) | - | 34 | 30 | 5 (5) | 0 | 5 (5) | 1, 2, 3 | 117 |
| 14 | 8 (8) | 1 | 8 (8) | 0 | 25 | 31 | 3 (3) | 0 | 2 (2) | 1, 2 | 77 |
| 15 | 0 (0) | - | 4 (4) | 1, 2 | 31 | 32 | 11 (10) | 1, 2 | 11 (10) | 1, 2 | 131 |
| 16 | 8 (0) | - | 8 (0) | - | 78 | 33 | 4 (4) | 1, 2 | 0 (0) | - | 19 |
| 17 | 4 (4) | 0 | 4 (4) | 2 | 60 | | | | | | |

## Appendix B. Segmentation of New Fluids

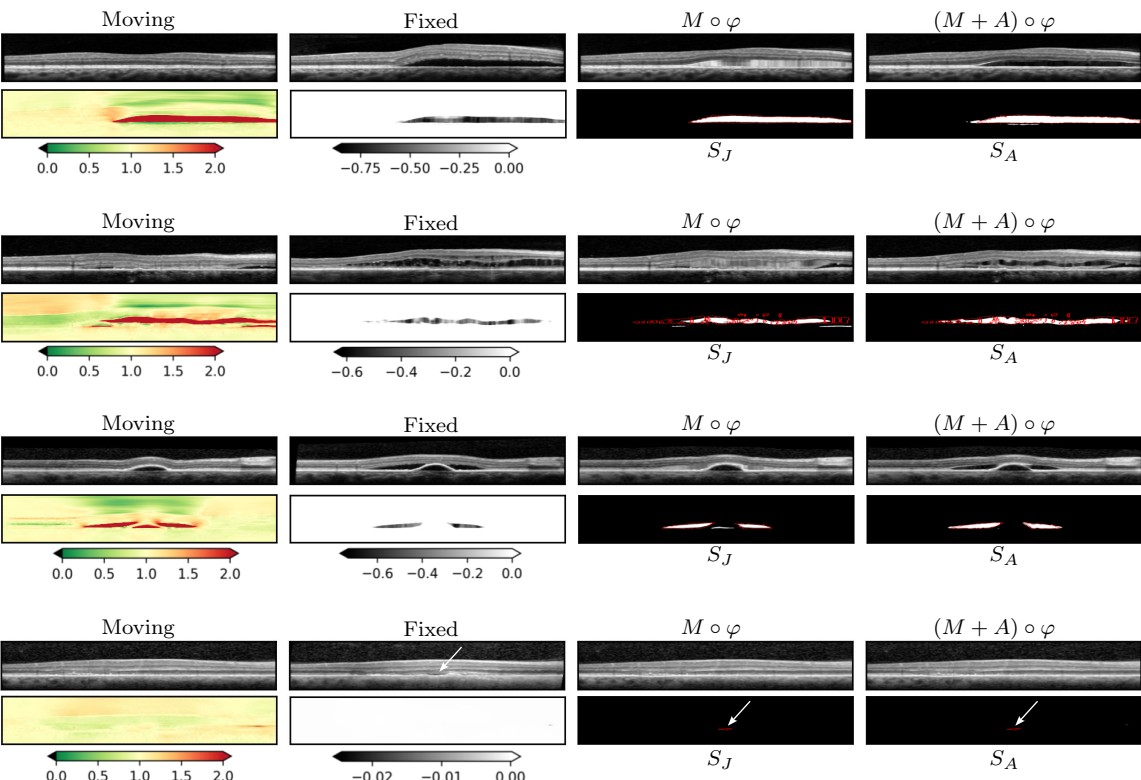

Figure 5: Four examples of new fluid segmentations resulting from our registration frame-work. New lesions are found with thresholding on the Jacobian determinant of the deformation field and on the deformed appearance offsets. For each patient, in the top row, we show the moving and fixed images and the deformed moving image with and without appearance offsets. In the lower row, the Jacobian determinant of the deformation, the deformed appearance map and the resulting segmentations are shown. Ground truth lesion segmentations are overlaid in red. The top example is the best-functioning image with a DSC of 0.94 and the bottom is an example of a fluid that is not detected (white arrow).

