# OpenReview forum: "FluidRegNet: Longitudinal registration of retinal OCT images with new pathological fluids"
_MIDL.io/2024/Conference — MIDL 2024 Oral_

### Official Review · Reviewer_NqkW · 2024-02-28

**Confidence:** 4
**Preliminary Rating:** 5
**Final Rating:** 5

**Summary:**

This work presents a method for OCT registration that attempts to address the presence of spontaneously occurring or disappearing fluid deposits. This is achieved by detecting fluid deposits in the images, and injecting a small fake fluid deposit in the paired image at the corresponding spot. By registering the adapted pairs, the small fake fluid deposits can be blown up to match the dark image intensities in the true deposits, which prevents large errors in the deformations fields caused by registration methods looking for "a compromise" of dark image intensities to fill the fluid deposits. This approach additionally enables unsupervised fluid segmentation.

**Strengths:**

The metamorphosis model is very suitable for the task at hand. The reliance on rough threshold-based segmentations is a clever way of avoiding the need for high-quality segmentations.

The paper is well written; the explanations are generally easy to follow.

The performance of the method seems to be very good, both in terms of registration accuracy and for unsupervised fluid segmentation.

**Weaknesses:**

Most of the figures are too small to see. This makes it unpleasant to read the printed version of the paper.

The authors state that they trained the network using OCT scans from 61 eyes, but there is no information on what images were used for evaluation.

**Detailed Comments:**

There are many irrelevant significant digits in table 2, which makes it messy and harder to read. It would be better to round these numbers to two decimal places.

**Justification Of Final Rating:**

My justification remains the same as for my preliminary rating: the paper presents a sensible, well executed method with a relevant methodological contribution. The concerns I expressed have been properly addressed, and I have no further objections to accepting the paper.

**Justification Of The Preliminary Rating:**

While the figures could use some significant attention to make them easier to read, the paper presents a sensible, well executed method with a relevant methodological contribution. My only worry is whether the evaluation was properly done, as information on the evaluation seems to be missing from the current version of the paper. This should be added in the rebuttal.

**Questions To Address In The Rebuttal:**

The information on what images were used for evaluation is missing. This should be added.

---

> ### Author Response · Authors · 2024-03-12
>
> Thank you very much for reviewing our paper and your helpful remarks.
>
> We agree that the figures are rather small and enlarged them in the new version of the paper. We also revised Table 2 to contain only two decimal places.
>
> Regarding the evaluation of our method, you are right that we missed explaining which data is used for evaluation. A five-fold cross-validation has been used, where the images of 80% of the patients have been used for training and the remaining images for testing. Results have been calculated over all folds and finally averaged over all test images. This description has been added to the paper.

---

### Official Review · Reviewer_MBaQ · 2024-02-28

**Confidence:** 4
**Preliminary Rating:** 4
**Recommendation:** Poster
**Final Rating:** 4

**Summary:**

This paper discusses the problem of missing correspondences for the registration of images with newly emerging fluids in the retina, which can severely impair a patient's vision. The paper proposes a registration framework called FluidRegNet that addresses this issue by mimicking the onset of fluid formation in the moving image by inserting minimal regions of changed intensities called lesion appearance seeds. The framework assumes that the deformations inside and outside of fluid deposits are different and uses a masked loss function to train the network. The correct quantification of fluid development is crucial in understanding disease development and progression.

**Strengths:**

This paper presents a novel approach for fluid segmentation in longitudinal retinal OCT scans.The proposed method handles temporally varying fluid types, which is difficult. Their approach, FluidRegNet outperformed the other approaches in the experiment.

**Weaknesses:**

One weakness of the paper is that it does not provide a good novelty in the architecture of the proposed method. While the paper mentions that the method is fully convolutional and uses a U-Net-like architecture, it does not provide a detailed description of the network's layers or parameters. Thus, their outperformance may be from the design of a new loss function.

Another weakness of the paper is that it does not provide a detailed analysis of the limitations of the proposed method.

**Detailed Comments:**

This paper is well-written and presents a thorough evaluation of the proposed method. The paper could benefit from a more detailed discussion of the clinical implications of the proposed method. While the paper mentions that the method could be used for the diagnosis and monitoring of retinal diseases, it does not provide a detailed analysis of the potential benefits or drawbacks of using the method in clinical practice. Providing this information would be valuable for clinicians and researchers who are interested in adopting the method for clinical use.

**Justification Of Final Rating:**

This paper discusses the problem of missing correspondences for the registration of images with newly emerging fluids in the retina, which can severely impair a patient's vision. The paper proposes a registration framework called FluidRegNet that addresses this issue by mimicking the onset of fluid formation in the moving image by inserting minimal regions of changed intensities called lesion appearance seeds. The framework assumes that the deformations inside and outside of fluid deposits are different and uses a masked loss function to train the network. The correct quantification of fluid development is crucial in understanding disease development and progression.

By adding more description on the network architecture and the limitations of the method, I believe that this draft became more solid candidate for a possible accepted paper. I will sustain my initial evaluation, which was '4: Weak accept.'

**Justification Of The Preliminary Rating:**

One weakness of the paper is that it does not provide a good novelty in the architecture of the proposed method. While the paper mentions that the method is fully convolutional and uses a U-Net-like architecture, it does not provide a detailed description of the network's layers or parameters. Thus, their outperformance may be from the design of a new loss function.

Another weakness of the paper is that it does not provide a detailed analysis of the limitations of the proposed method.

However, this approach will be applied to the registration of temporally varying retinal images, which is challenging. Thus, I recommend a weak accept for this paper.

**Questions To Address In The Rebuttal:**

Detailed information on the model architecture with the number of layers or nodes will supplement this paper. Also, more details in designing loss functions will strengthen the paper.

---

> ### Author Response · Authors · 2024-03-12
>
> Thank you very much for the review of our paper and your detailed remarks.
>
> Firstly, we would like to comment on your remark about the missing novelty in the network architecture. The main novelty of our method lies in the sparse fluid seed approach and the training procedure to achieve meaningful appearance offsets (combination of loss functions and 3-step training). Network design has not been the focus of our work, which is why we used an established network architecture, with only slight modifications, namely separate input layers and output heads. Nevertheless, the gain in performance that can be attributed to the network architecture (compared to Voxelmorph) is considerable, as can be seen from the results in Table 1 comparing the “classical” version of FluidRegNet to Voxelmorph since both networks are trained with the same loss functions. In view of the limited space available, we have decided to leave the illustration of the detailed network architecture in the appendix but have expanded the network description in the main paper.
>
> Thank you for your suggestion to add a more detailed analysis of the limitations of the method. We implemented this in the discussion section.
>
> Upon your remark we added a justification of the choice of the individual loss components in the paper. Another aspect that might not have been fully clear is how to find the weighting parameters in the loss function. This is indeed one of the major obstacles for adapting our method to different settings or registration problems since the weighting parameters need to be well balanced to assure a good registration performance and appearance seeds that are neither too small nor too big. We recommend finding the parameters by training FluidRegNet as a classical image registration network first using only the image distance and a spatial regularizer (evaluated on the entire image domain). Once you have found good hyperparameters for these, you can perform the pre-training step. The next step is the warming-up phase in which only the appearance output head is trained. This step is well suited to find a good weighting parameter for the sparsity loss. In all our experiments, the weighting parameters for the Dice loss and the Jacobian have proven less crucial, they only need to be “high enough” to give guidance to the registration and avoid inversions. A short version of this description has been added to the discussion section.

---

> > ### Comment · Reviewer_MBaQ · 2024-03-27
> > **Sustain my initial evaluation**
> >
> > By adding more description on the network architecture and the limitations of the method, I believe that this draft became more solid candidate for a possible accepted paper. I will sustain my initial evaluation, which was '4: Weak accept.'

---

### Official Review · Reviewer_zzpJ · 2024-02-29

**Confidence:** 4
**Preliminary Rating:** 5
**Recommendation:** Oral
**Final Rating:** 5

**Summary:**

The paper addresses the challenge of capturing the dynamics of retinal fluid growth in conditions like chronic central serous chorioretinopathy (CSCR), where fluid deposits alter the retina and impair vision. Previous methods often require segmentation and prior knowledge of lesions, which can be limiting. The proposed approach utilizes a novel sparse appearance seed method to model emerging fluids and lesion growth/shrinkage without the need for ground truth labels in pathological regions during training.

**Strengths:**

- Good overview of SOTA

- Interesting approach based on metamorphosis models to jointly estimate appearance and deformation field.

- Very good results, equivalent to SOTA for healthy tissue and above SOTA for pathological lesions.

**Weaknesses:**

- Training involves three steps: initially focusing on deformation output, then generating sparse appearance while keeping deformation frozen, and finally fine-tuning the entire network. What about the computational cost vs the other methods?

**Detailed Comments:**

- In Table 1: All images are registered to the subsequent follow-up image? Or sometimes certain image are skipped?

- Also in Table 1, it would be good to report the confidence intervals, or the standard deviation

- In general, what is the proportion of lesions in the dataset?

**Justification Of Final Rating:**

Based on the authors response my points were properly addressed and some confusing sentences in the paper were rephrased. I don't have any further comments. I think it is a good paper, so I don't have any reason to change my rating.

**Justification Of The Preliminary Rating:**

Sound paper, with comparisons to both deep learning approaches and classic medical registration tools. The method is interesting, it tackles an important issue like the appearance of lesions in longitudinal data and how to register previous healthy images and also track the emerging lesions. Moreover, they achieve SOTA in the registration of healthy tissue, i.e: they don't sacrifice performance on that.

**Questions To Address In The Rebuttal:**

- In the last part, it is discussed the potential of the method to estimate the unsupervised fluid segmentation in subsequent time-points if an initial segmentation is provided. Nevertheless, given the way the segmentation are discarded and that the fluid is 'already there', not appearing, I feel like this could be applied also with classical approaches, like SyN, why is it a 'strength' of the model then?

**Special Issue:**

No

---

> ### Author Response · Authors · 2024-03-12
>
> Thank you very much for reviewing our paper and your valuable feedback.
>
> Regarding your point about the computational cost of our method, it is true that FluidRegNet requires more storage than Voxelmorph and takes longer for training due to the larger number of parameters. Still, with a memory consumption of less than 5 GBs of GPU RAM during training (when using a batch size of 10), FluidRegNet does not require more storage than other U-Net-based CNNs and can be trained on a single GPU. With the significant performance gain compared to Voxelmorph, we consider the difference in storage requirements (and training times) justifiable. Additionally, we would like to mention that we used 700 training epochs for all image registration networks, thus the number of training steps is the same for all methods compared. In the paper, we have extended the description of network training both for FluidRegNet and the competitive methods.
>
> Yes, in Table 1 results are reported for registering each image to its subsequent follow-up image. We added this to the paper and revised Table 1 to show the standard deviations of the considered metrics.
>
> Upon your remark, we estimated the proportion of lesions in the dataset based on the images with given manual annotations. Out of 163 manually annotated OCT volumes, 105 contained lesions (11x IRF – 66 B-scans, 78x SRF – 715 B-scans and 65x PED – 280 B-scans). This description has been added to the paper. We would also like to draw your attention to Table 3 in the appendix, which provides a more detailed breakdown of which biomarkers are present in which patients.
>
> Finally, we would also like to comment on your remark about using FluidRegNet for unsupervised fluid segmentation in subsequent time points. Given the nature of CSCR, we consider a classic method to be unsuitable. Fluids do indeed (dis-)appear between time points; Figure 3 shows patients for whom this is the case. The initial segmentation of the very first time point is thus only needed for persistent fluids that are seldom throughout our dataset. For more clarity, we reformulated the respective sentences in Section 3.3.2.

---

### Meta-Review · Area_Chair_sg2e · 2024-04-03

**Recommendation:** Accept (Oral)
**Confidence:** 5

**Metareview:**

All reviewers agreed that addressing the problem of longitudinal registration of retinal OCT images is an important and challenging task in medical image analysis. The proposed approach, which involves jointly estimating appearance and geometric deformations in time-series OCT images, is interesting and under-explored in the literature. Given the consistent positive recommendations from all reviewers, the meta reviewer is pleased to recommend acceptance of the paper. However, the authors are strongly encouraged to thoroughly address all comments raised by the reviewers in their final revisions.

---

### Decision · Program_Chairs · 2024-04-05

Accept (Oral)